# Tunable rectification in a molecular heterojunction with two-dimensional semiconductors

Jaeho Shin[1,4], Seunghoon Yang [1,4], Yeonsik Jang[2], Jung Sun Eo[1], Tae-Wook Kim [3], Takhee Lee [2], Chul-Ho Lee [1✉] & Gunuk Wang[1✉]

Until now, a specifically designed functional molecular species has been recognized as an absolute necessity for realizing the diode's behavior in molecular electronic junctions. Here, we suggest a facile approach for the implementation of a tailored diode in a molecular junction based on non-functionalized alkyl and conjugated molecular monolayers. A two-dimensional semiconductor ($MoS_2$ and $WSe_2$) is used as a rectifying designer at the alkyl or conjugated molecule/Au interface. From the adjustment of band alignment at molecules/two-dimensional semiconductor interface that can activate different transport pathways depending on the voltage polarity, the rectifying characteristics can be implemented and controlled. The rectification ratio could be widely tuned from 1.24 to $1.83 \times 10^4$ by changing the molecular species and type and the number of layers of the two-dimensional semiconductors in the heterostructure molecular junction. Our work sets a design rule for implementing tailored-diode function in a molecular heterojunction structure with non-functionalized molecular systems.

[1] KU-KIST Graduate School of Converging Science and Technology, Korea University, 145 Anam-ro, Seongbuk-gu, Seoul 02841, Republic of Korea. [2] Department of Physics and Astronomy, and Institute of Applied Physics, Seoul National University, Seoul 08826, Republic of Korea. [3] Department of Flexible and Printable Electronics, Jeonbuk National University, Baekje-daero 567, Deokjin-gu, Jeonju 54896, Republic of Korea. [4] These authors contributed equally: Jaeho Shin, Seunghoon Yang. ✉email: chlee80@korea.ac.kr; gunukwang@korea.ac.kr

The ultimate objectives in the field of molecular electronics are to realize electronic functionalities, such as rectifying, optical switching, and thermoelectric effects, at the device miniaturization limit and to systemize all the aspects of the charge transport mechanisms for rational device design[1–12]. Specially designed molecular species have been utilized for the realization of specific (or desired) device functions. For example, the D-σ-A type molecules have been widely used for a diode component, where the highest-occupied molecular orbital (HOMO) of the donor and the lowest-unoccupied molecular orbital (LUMO) of the acceptor are closely aligned to the Fermi level of electrodes, i.e., the frontier orbital level of each molecular unit is asymmetrically positioned with respect to the electrodes[9,10]. This asymmetric energy alignment in the junction can cause the different sequential tunneling through the acceptor and donor depending on the polarity of the applied voltage, which could yield the rectification property of the molecular junction. In the case of the ferrocenyl molecules that composed of long alkyl chain and ferrocenyl termini unit, they can also exhibit the rectification characteristic because the charge transport pathways are mainly determined by the relative HOMO level of the ferrocenyl unit according to voltage polarities[11,12]. In this sense, specially designed molecular species that can differently engineer the band alignment of the molecular units according to the voltage polarity is dispensable for implementing a desirable diode's behavior in molecular electronic junction.

In this point of view, non-functionalized molecules such as alkanethiol or conjugated molecules that only exhibit symmetric tunneling transport have been excluded from the realization of electronic diode functions[13,14]. However, even these widely studied molecules can present the desired electronic functionality as the interfacial band alignment is properly adjusted in molecular heterojunctions. Here, we present a novel strategy and design rule for realizing molecular-scale diode features based on the energy band engineering between simple alkanethiol or conjugated molecules and 2D semiconductors. We systematically engineer the band alignment by inserting 2D semiconductors between molecules and metal (Au), resulting in different charge transport pathways according to voltage polarities without introducing specially designed molecules. In addition, the rectifying characteristics are tunable by controlling the essential constituents such as the molecular species, molecular length, 2D semiconductors, and the number of layers. Our work sets a generalized design rule for implementing rectifying characteristics in a molecular heterojunction structure with non-functionalized molecules and 2D semiconductors.

## Results

**Molecular heterojunction structure.** Figure 1a shows a schematic diagram of the molecular heterojunction structure composed of a molecular self-assembled monolayer (SAM) and $MoS_2$ stacked on an $Au/SiO_2/Si$ substrate, of which the electrical properties are investigated by conductive atomic force microscopy (CAFM). Five non-functionalized molecular species that differ in terms of the molecular length and the HOMO-LUMO gap (i.e., benzene-1-monothiol (denoted as OPT1), biphenyl-4-monothiol (OPT2), 1-octanemonothiol (C8), 1-decanemonothiol (C10), and 1-dodecanemonothiol (C12)) are used herein. As a representative 2D semiconductor, an $n$-type $N_L$-$MoS_2$ with different numbers of layers ($N_L = 1_L$, $2_L$, and $3_L$) and $p$-type $1_L$-$WSe_2$ are used to form a heterojunction with the SAMs (right of Fig. 1a). Introducing 2D semiconductors (sub-1 nm thick) at the molecule/Au interface allows us to modify the interfacial band profile across the junction while maintaining the molecular-scale junction size. In addition, judicious choice of the 2D semiconductor type ($n$- or $p$-type) and the number of semiconductor layers enables adjustment of the interfacial barrier between the SAMs and 2D semiconductors, as well as the majority carriers, in a highly designed manner. Figure 1b shows the optical images and topological line profiles of $MoS_2$ and $WSe_2$ exfoliated on the $SiO_2/Si$ substrate. The measured height of ~0.7 nm indicates that both $MoS_2$ and $WSe_2$ are monolayer thick. In the Raman spectra shown in Fig. 1c, the $E^1_{2g}$ and $A_{1g}$ vibrational modes are observed at 383 and 402 $cm^{-1}$ with of spacing of $\Delta = 18.5\ cm^{-1}$ for monolayer ($1_L$)-$MoS_2$ (red line) and 247 $cm^{-1}$ with spectral

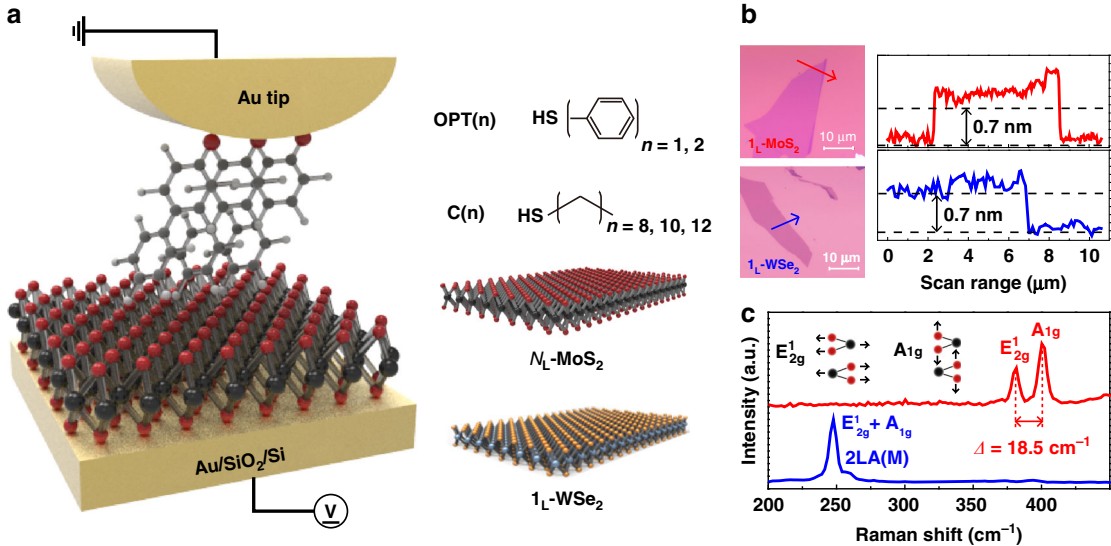

**Fig. 1 Molecular heterojunction structure. a** Schematic of molecular heterojunction composed of OPT2 and $1_L$-$MoS_2$ stacked on an $Au/SiO_2/Si$ substrate using the CAFM technique (left). Note that the tip-loading force ($F_L$) was fixed at 1 nN to prevent variation in the interfacial coupling between the molecules and TMDs. Five different molecular species (OPT ($n = 1, 2$) and C ($n = 8, 10, 12$)) and two different 2D semiconductor types ($N_L$ ($1_L$, $2_L$, or $3_L$)-$MoS_2$, and $1_L$-$WSe_2$) are shown (Right). **b** Topological line-profiles of $1_L$-$MoS_2$ (red line) and $1_L$-$WSe_2$ (blue line) on $SiO_2/Si$ substrate from AFM in non-contact scanning mode. Arrow in optical image of each TMD indicates the investigation range of the line-profiles. **c** Raman spectra of $1_L$-$MoS_2$ (red line) and $1_L$-$WSe_2$ (blue line).

overlapping for $1_L$-WSe$_2$ (blue line), confirming the monolayer structure[15,16]. AFM investigation, Raman, and photoluminescence (PL) spectroscopy are used to characterize the various transition metal dichalcogenides (TMDs) used in this work (Supplementary Fig. 1)[15,16]. For the electrical measurements, the Au tip coated with the non-functionalized SAMs is carefully placed on the 2D semiconductors with the loading force $(F_L) = 1$ nN[17]. The details of the experimental methods and sample preparation are described in the Methods section.

**Electrical characteristics**. Figure 2a shows the representative current-voltage (I-V) characteristics for five different molecular junctions, i.e., Au/OPT2/$1_L$-MoS$_2$/Au (red solid circle), Au/OPT2/$1_L$-WSe$_2$/Au (cyan solid circle), Au/OPT2/Au, Au/$1_L$-WSe$_2$/Au, and Au/$1_L$-MoS$_2$/Au. Note that the Au tip is grounded, and a voltage is applied to the bottom Au electrode. For the junctions composed of only TMDs or OPT2 (represented by black dashed lines), all I-V characteristics exhibit symmetric behavior (rectification ratio (RR) = ~1, defined as $|I (V = 1 \text{ V})|/|I (V = -1 \text{ V})|$) due to the single transport barrier located in-between the Fermi level ($E_F$) of the Au electrodes. However, when the molecular heterojunctions is formed with $1_L$-MoS$_2$ and $1_L$-WSe$_2$, the I-V curves change asymmetrically. In particular, the RR of these devices is strongly dependent on the 2D semiconductor type. For the OPT2/$1_L$-MoS$_2$ junction, RR = $1.79 \times 10^3$, which is much larger than that of the OPT2/$1_L$-WSe$_2$ junction (RR = 2.31) (Fig. 2a). In fact, this value exceeds the theoretical rectification limit (RR = 20) of a molecular junction containing only σ- or π-bonded molecular SAMs[18]. More specifically, the OPT2/$1_L$-MoS$_2$

junction shows distinct asymmetry compared with the OPT2/$1_L$-WSe$_2$ junction, with a lower I at $V < 0$ and higher I at $V > 0$. Such rectifying behavior is reproducibly obtained for both molecular heterojunctions through statistical investigation of the I-V characteristics on several positions (more than five positions) at different samples (more than four samples) (200–1100 times), as shown in the contour plots in Fig. 2b and Supplementary Figs. 2–4. The statistical histogram of RR for the OPT2/$1_L$-MoS$_2$ junction shows a higher value (RR = $(1.38 \pm 0.73) \times 10^3$) than that for the OPT2/$1_L$-WSe$_2$ junction (RR = $2.46 \pm 1.42$) (Fig. 2c). Note that the maximum RR for the OPT2/$1_L$-MoS$_2$ junction is found to be $\sim1.83 \times 10^4$ (Supplementary Fig. 4).

**Interfacial energy band alignment for molecular heterojunction**. To understand the rectifying mechanism and the difference in the RR for the OPT2/$1_L$-MoS$_2$ and OPT2/$1_L$-WSe$_2$ junctions, we establish different interfacial energy band alignments for both heterojunctions, corresponding to the different applied voltages ($V = 0$, 1.0, and −1.0 V) (Fig. 2d and Supplementary Fig. 5). For example, at $V = 0$ V, the equilibrium energy band diagrams of the molecular heterojunctions can be schematically represented by considering the $E_F$ position of Au and the energy bands of the interfacial constituents (OPT2 and TMDs). Three interfaces are consistently present in the molecular heterojunctions: top Au-tip/OPT2, OPT2/TMDs, and TMDs/bottom Au electrode. Because the interface of top Au-tip/OPT2 is chemically bonded by Au-S covalent contact, the OPT2 orbital levels having a HOMO-LUMO gap ($E_g$) = ~4 eV could shift in response to the change in the $E_F$ of the top Au-tip upon application of a voltage[14,17]. At the

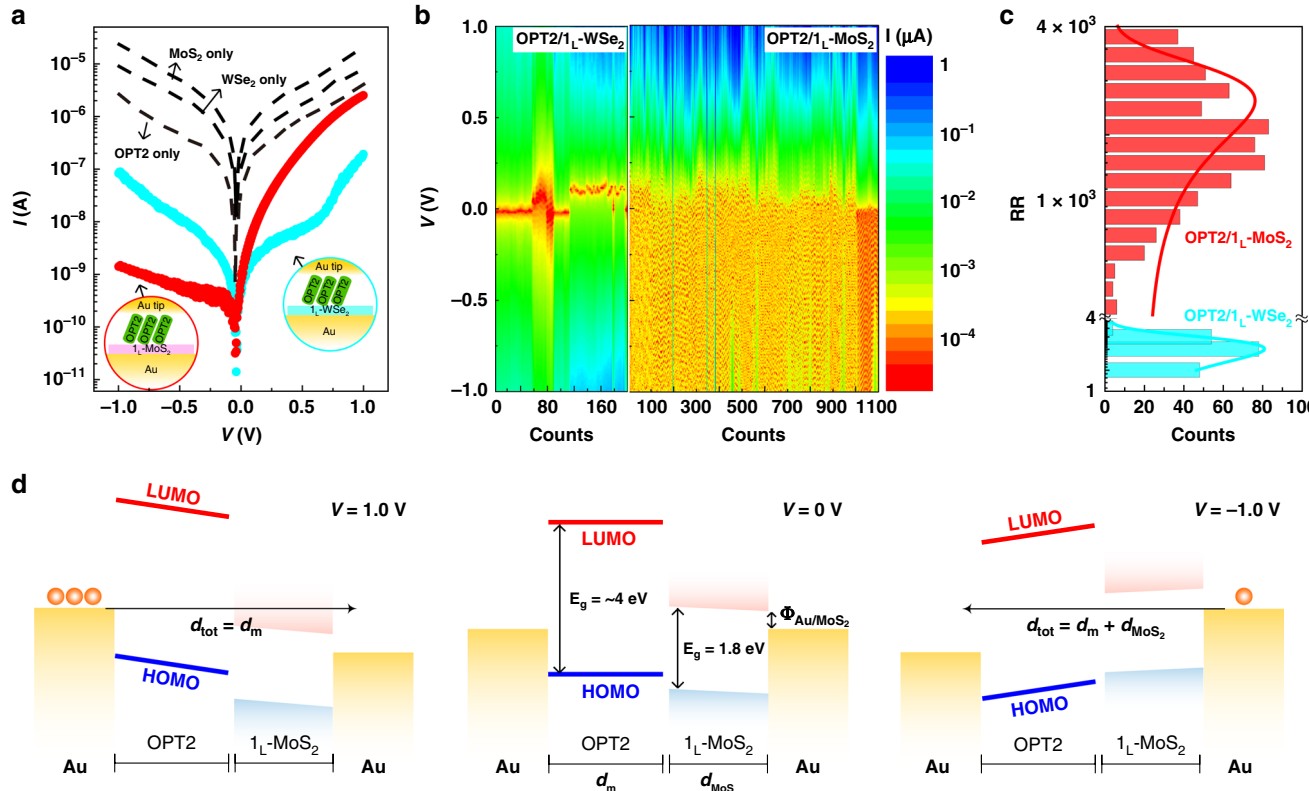

**Fig. 2 Rectifying characteristics and energy band alignment of molecular heterojunction. a** Representative I-V characteristics of the junctions without OPT2 or TMDs (black dotted line), OPT2/$1_L$-WSe$_2$ (cyan solid circle), and OPT2/$1_L$-MoS$_2$ (red solid circle) junctions. **b** Contour maps of transport I for OPT2/$1_L$-WSe$_2$ (left) and OPT2/$1_L$-MoS$_2$ (right) junctions according to V and number of junctions. **c** Statistical histograms of RR for OPT2/$1_L$-MoS$_2$ and OPT2/$1_L$-WSe$_2$ junctions. The line curves are fitting results from Gaussian function. Note the total numbers of OPT2/$1_L$-WSe$_2$ and OPT2/$1_L$-MoS$_2$ junctions are 200 and 1100, respectively. **d** Interfacial energy band alignments for Au/OPT2/$1_L$-MoS$_2$/Au junction at $V = 0$ V, $V = 1.0$ V, and $V = -1.0$ V. Note that the $E_g$ of $1_L$-MoS$_2$ is set to ~1.8 eV.

interface of OPT2/TMDs, however, the TMD layers cannot undergo chemical-linking with the anchoring group (–CH) of OPT2 on one side. When a voltage is applied, the interfacial barrier between OPT2 and the TMDs changes. Lastly, at the interface of TMDs/bottom Au, the $E_F$ of the bottom Au electrode is strongly pinned to the bandgap of the TMDs due to the dichalcogenide vacancy energy states[19–22], forming a constant interfacial TMD/Au barrier even when a voltage is applied. Under this circumstance, the energy band alignments at the Au-tip/ OPT2 and TMD/Au interfaces can shift independently according to the applied voltage. Consequently, the effective transport width could be varied depending on the voltage polarity and the TMD type (Fig. 2d and Supplementary Fig. 5). For example, in the case of OPT2/$1_L$-MoS$_2$, the conduction band edge of $1_L$-MoS$_2$ is located between the applied bias window when $V = 1.0$ V, which eventually shortens the effective transport width ($d_{tot} = d_m$) (the left of Fig. 2d). This interfacial energy band alignment significantly enhances the transport because the majority carriers (i.e., electrons in this case) are transported only across the OPT2 barrier ($d_{tot} = d_m$), and not across the other interfacial barriers. In contrast, at $V = -1.0$ V, the transport can be largely suppressed by the longer transport width ($d_{tot} = d_m + d_{MoS_2}$) because of the additional $1_L$-MoS$_2$ barrier (the right of Fig. 2d). As a result, this heterojunction type can lead to a larger RR. In the case of OPT2/ $1_L$-WSe$_2$, however, the majority carriers (i.e., holes in this case) must move across all the interfacial barriers (both the OPT2 and the $1_L$-WSe$_2$ barriers) regardless of the voltage polarities (Supplementary Fig. 5). This is because the valance band edge of $1_L$-WSe$_2$ could not be located between the applied bias window due to the midgap pinning[21,22], leading to a smaller RR. In order to further investigate the Fermi-level pinning behavior at the metal/ WSe$_2$ interface, we additionally conduct the $I$-$V$ measurements for the OPT2/$1_L$-WSe$_2$ junction using another bottom electrode (Pt) in addition to Au. The electrical characteristic and the extracted RR value are rarely changed as compared with those of the OPT2/$1_L$-WSe$_2$ junction using the Au electrode (Supplementary Figs. 6 and 7). This is presumably because the Fermi-level of the metal is similarly pinned at the mid-gap states of $1_L$-WSe$_2$ regardless of the metal work functions (Au (~5.1 eV) and Pt (5.7 eV)) although the Fermi levels of both metals can align relatively well with the valance band edge of WSe$_2$ only if considering their work functions[23]. According to previous literature, it has been well known that the mid-gap states can be formed by both defect states originated from naturally existing vacancies of WSe$_2$ and virtual gap states induced by metal wave function penetration, which results in the strong Fermi-level pinning[22]. Therefore, our control experimental results support our energy-band diagram model for the Au/OPT2/$1_L$-WSe$_2$/Au junction that is based on the mid-gap pining for the WSe$_2$ case. It is noted that an additional experiment such as the analysis of gate-dependent electrical characteristics is desired for the direct evidence for where the Fermi-level of the electrode is aligned into the $1_L$-WSe$_2$. Considering these facts, the interfacial band alignments and the change in the transport widths depending on the voltage polarity are recognized as important factors influencing the rectifying properties in these molecular heterojunctions.

**Tunable rectification in molecular heterojunction.** Engineering of the molecular rectifying features by controlling the essential constituents of the molecular heterojunction, such as the molecular species, molecular length, and the number of MoS$_2$ layers, is investigated. Note that the statistical data for molecular SAMs/$1_L$-MoS$_2$ junctions in Fig. 3 are obtained from $1_L$-MoS$_2$ (Sample #1). Figure 3a shows the representative $I$-$V$ characteristics for the Au/ OPT2 (or C12)/$N_L$-MoS$_2$ (number of layer ($N_L$) = $1_L$, bilayer ($2_L$),

or trilayer ($3_L$))/Au junction. Three interesting transport phenomena are observed. First, $I$ is approximately three orders of magnitude higher for the OPT2/$N_L$-MoS$_2$ junction than for the C12/$N_L$-MoS$_2$ junction. The degree of decay in the quantum wave function through the OPT2 tunnel barrier is smaller than that through the C12 barrier due to the smaller HOMO-LUMO gap and the shorter molecular length, leading to greater conductance in OPT2/$N_L$-MoS$_2$ junction. Second, in the forward-bias region, $I$ remains almost constant, independent of the number of MoS$_2$ layers. In fact, because the conduction band edges of $N_L$-MoS$_2$ are located between the $E_F$ of both Au electrodes at $V > 0$, the transport $I$ is mainly determined by the molecular barrier height, not by the number of the MoS$_2$ layers. Lastly, in the reverse bias region, $I$ increases as the number of MoS$_2$ layers increases. In general, increasing the number of MoS$_2$ layers can lead to a reduction of the interfacial barrier height at the $N_L$-MoS$_2$/Au junction due to bandgap reduction derived from the strong interlayer electronic coupling between the sulfur atoms and the quantum confinement effect[24,25]. Furthermore, because charge transport at the Au/$N_L$-MoS$_2$/Au junction can be explained by the Schottky emission mechanism (Supplementary Fig. 8), the increase in the reverse bias current mainly originates from reduction of the MoS$_2$ barrier. Consequently, RR decreases with increasing number of MoS$_2$ layers. Figure 3b presents the statistical values (left) and histograms (right) of the RR for the OPT2/ $N_L$-MoS$_2$ and C12/$N_L$-MoS$_2$ junctions according to the number of MoS$_2$ layers. Generally, the RR decreases as the number of MoS$_2$ layers increases, regardless of the molecular species. Note that the same tendency is also observed for the other molecular heterojunctions (OPT1, C8, and C10-based) (Supplementary Figs. 9 and 10). This result implies that the use of $1_L$-MoS$_2$ maximizes the rectifying ability of the molecular heterojunction, regardless of the molecular species.

Figure 3c shows the representative $I$-$V$ characteristics for the Au/OPT($n$) ($n = 1$ or 2)/$1_L$-MoS$_2$/Au and Au/C($n$) ($n = 8$, 10, or 12)/$1_L$-MoS$_2$/Au junctions. Generally, longer molecules lead to decreased conductance due to the increase of the tunneling length in the molecular heterojunction[13,14]. However, the transport $I$ at $V < 0$ decreases more significantly than that at $V > 0$ when a longer molecule is used, leading to a larger RR. Inserting the thicker insulating layer between the metal and the semiconductor can further alleviate the $E_F$ pinning effects by reduction of the metal-induced gap state (MIGS) originating from further attenuation of the charge wave function across the insulating layer[26,27]. In this regard, inserting the longer molecules acting as an insulating layer between the Au-tip and $1_L$-MoS$_2$ might lead to further unpinning of the $E_F$ by reduction of MIGS density in $1_L$-MoS$_2$. As a result, the upward shift of the $1_L$-MoS$_2$ barrier due to the unpinning effect can further decrease $I$ at $V < 0$. Figure 3d presents the statistical average values (left) and histograms (right) of RR for the OPT($n$) (or C($n$))/$1_L$-MoS$_2$ junction according to the molecular length. As shown in Fig. 3d, RR increases when longer molecules are used, regardless of the number of MoS$_2$ layers. In particular, the RR of OPT2 is much larger than that of C8 despite the similar molecular length, which is due to the higher forward-bias current, attributed to the smaller HOMO-LUMO gap of the former[13,18]. This result allows us to define another design rule stating that molecules with longer backbone structures and smaller HOMO-LUMO gaps can improve the rectifying ability of the molecular heterojunction.

**Pathway-dependent charge transport mechanism and rectifier design rule.** In order to generalize the rectifying characteristics in the developed molecular heterojunctions, the electrical characteristics are theoretically modeled based on the pathway-dependent

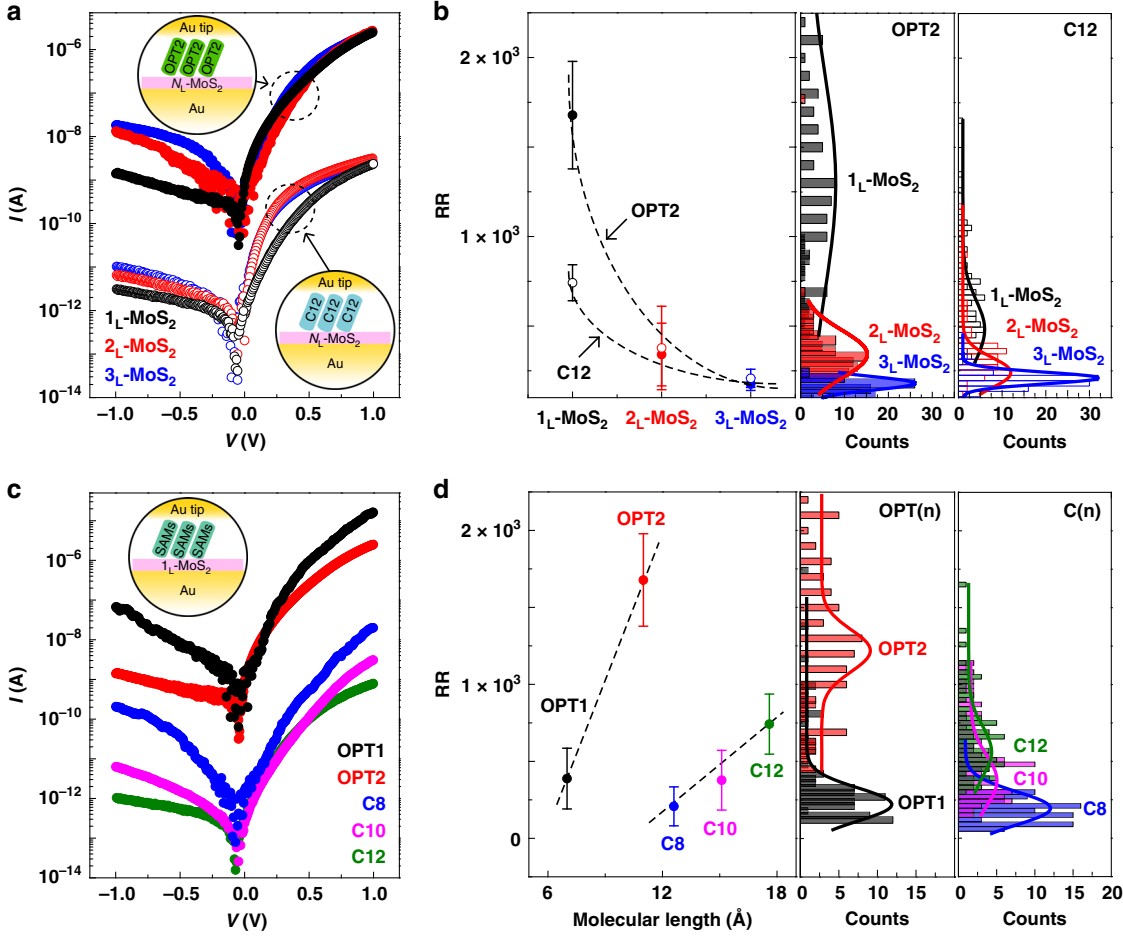

**Fig. 3 Modulation of rectifying characteristics of the molecular heterojunctions. a** Representative $I$-$V$ characteristics for the Au/OPT2 (or C12)/$N_L$-MoS$_2$ ($N_L = 1_L$, $2_L$, or $3_L$)/Au junction. **b** RR plots of OPT2/$N_L$-MoS$_2$ and C12/$N_L$-MoS$_2$ junctions as a function of the number of MoS$_2$ layers. As the $N_L$ was increased from 1 to 3, the RR decreased from $(1.68 \pm 0.63) \times 10^3$ to $(1.75 \pm 0.35) \times 10^2$ for the OPT2/$N_L$-MoS$_2$ junction, and from $(7.41 \pm 1.00) \times 10^2$ to $(2.07 \pm 0.17) \times 10^2$ for the C12/$N_L$-MoS$_2$ junction. The statistical histograms of RR for OPT2/$N_L$-MoS$_2$ and C12/$N_L$-MoS$_2$ as a function of the number of MoS$_2$ layers are shown in the right figure. **c** Representative $I$-$V$ characteristics for OPT($n$) ($n = 1$ or 2)/$1_L$-MoS$_2$ and C($n$) ($n = 8$, 10, or 12)/$1_L$-MoS$_2$ junctions. As $n$ increased, the RR increased from $(3.87 \pm 1.95) \times 10^2$ to $(1.68 \pm 0.63) \times 10^3$ for OPT($n$)/$1_L$-MoS$_2$, and $(2.08 \pm 1.26) \times 10^2$ to $(7.41 \pm 1.00) \times 10^2$ for the C($n$)/$1_L$-MoS$_2$ junction. **d** RR plots of OPT($n$)/$1_L$-MoS$_2$ and C($n$)/$1_L$-MoS$_2$ junctions as a function of the molecular length. The statistical histograms of RR for OPT($n$)/$1_L$-MoS$_2$ ($n = 1$ and 2) and C($n$)/$1_L$-MoS$_2$ ($n = 8$, 10, and 12) junctions are shown in the right figure. The error bars in (**b**) and (**d**) indicate the standard deviations of RR obtained from at least 100 different positions of each junction. Note that the molecular length for each molecule is 0.7 (OPT1), 1.1 (OPT2), 1.26 (C8), 1.51 (C10), and 1.76 nm (C12), respectively.

charge transport mechanism by considering the voltage polarity. As shown in Fig. 2d, tunneling across the molecular barrier is the dominant transport mechanism at $V > 0$, while the tunneling and Schottky emission across the molecular and $N_L$-MoS$_2$ barriers are the dominant transport mechanisms at $V < 0$, which largely suppresses the transport $I$. The details of the charge transport equations are further discussed in the Methods Section and Supplementary Information (Supplementary Fig. 11). Considering the effect of the applied voltage polarity on the pathway-dependent charge transport mechanisms, the current density ($J$)-$V$ characteristics and RR of the Au/OPT($n$) ($n = 1$ or 2) or C($n$) ($n = 8$, 10, or 12)/$N_L$-MoS$_2$/Au junctions (Fig. 4a, b) are estimated. The estimated RR values are in good agreement with the experimental RR value, and the trend of RR depending on the junction constituents is also well described. Further, to establish a design rule for the diode characteristics, we predict the RR as a function of the molecular length and barrier height in non-functionalized molecular junction systems with interfacial $N_L$-MoS$_2$, as shown in the contour plots in Fig. 4c for molecular lengths from 0.5 to 2.5 nm and barrier heights from 1.5 to 6.0 eV. Note that the selected

range of molecular variables can cover the major σ- and π-bonded molecular SAMs that are widely used in molecular junctions[13,18]. In Fig. 4c, the RR increases with increasing molecular length and lower molecular barrier height, but decreases as the number of MoS$_2$ layers increases, consistent with the experimental results. From these RR contour maps, the rectifying characteristics of molecular heterojunctions consisting of non-functionalized SAMs and $N_L$-MoS$_2$ could be generalized and the molecular junction structure having the maximum RR could be designed.

In summary, we have first implemented a molecular rectifier composed of simple molecules and 2D semiconductors. The rectifying characteristics can be implemented and further tuned by engineering band alignment between non-functional molecules and 2D semiconductors which make different transport pathways according to voltage polarities. Given that non-functional SAM-based molecular heterojunctions employing 2D semiconductors exhibit strong rectifying characteristics and that their performance can be controlled by implementing the suggested design rule, a tailored molecular rectifier based on simple σ- and π-bonded SAMs can potentially be realized.

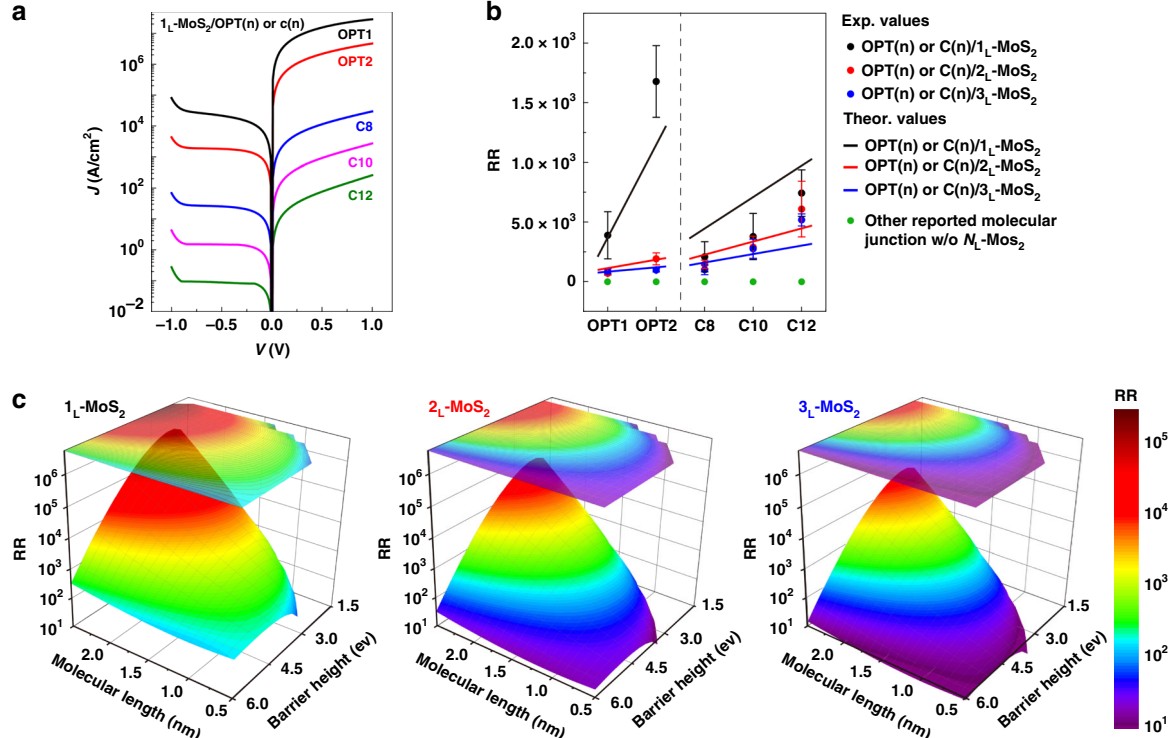

**Fig. 4 Generalization of molecular heterojunction rectifiers. a** $J$-$V$ plots for Au/OPT($n$) ($n = 1$ or 2) and C($n$) ($n = 8$, 10, or 12)/$1_L$-MoS$_2$/Au junctions calculated from the pathway-dependent charge transport mechanism (Eqs. (1)–(3) and Supplementary Fig. 11). **b** RR plots as a function of the molecular species. The solid lines were calculated from the pathway-dependent charge transport mechanism. The green solid circles are the RR values obtained from other reported molecular junctions consisting of simple $\sigma$- or $\pi$-bonded molecules without MoS$_2$ (RR = ~1). The error bars indicate the standard deviations of RR obtained from at least 100 different positions of each junction. **c** Contour plots of estimated RR as a function of the molecular length and barrier height for the molecular heterojunction with different numbers of MoS$_2$ layers ($N_L = 1_L$, $2_L$, and $3_L$).

## Methods

**Sample preparation and characterization.** All 2D semiconductors ($1_L$-/$2_L$-/$3_L$-MoS$_2$ and $1_L$-WSe$_2$) are prepared by using the typical mechanical exfoliation and transfer method. The exfoliated MoS$_2$ and WSe$_2$ layers are picked up by using a poly-propylene carbonate (PPC)/polydimethyl siloxane (PDMS) stamp and then mechanically transferred onto the Au/SiO$_2$/Si substrate. The number of TMD layers is verified by optical contrast, after which AFM in non-contact mode (Park NX10, Park Systems Corp., South Korea) is used to determine the thickness of each layer of the TMDs. The Raman and PL spectra of the TMDs are obtained using a home-built spectrometer equipped with a monochromator (Andor, SOLIS 303i) and a 532 nm excitation laser with a spot diameter of 0.5 μm. The signal is collected by an objective lens (100× NA = 0.9) and dispersed by 1200 and 300 line/mm gratings for the Raman and PL measurements. To form the densely packed non-functionalized SAMs on the Au tip, the tip is immersed in the molecular solutions (~5 mM ethanol) for several hours in a glove-box filled with nitrogen gas with less than 10 ppm O$_2$. To remove the non-assembled residual molecules from the Au tip, the tip is repeatedly rinsed with ethanol and was blown dry using N$_2$.

**Electrical characterization.** The Au tip coated with the non-functionalized SAMs is carefully placed at the bottom of the 2D semiconductor with $F_L = 1$ nN. For the electrical measurement, the Au tip is grounded, and a voltage is applied to the bottom Au electrode. The electrical $I$-$V$ characteristics are measured in stationary mode with $F_L = 1$ nN using a DLPCA-2000 built-in current amplifier (Electro-Optical Components) at a humidity of <15%. To obtain statistically meaningful data, the electrical measurements are repeated at different junction positions for each molecular junction (at least 100 times).

**Estimated number of contacted molecules.** For the OPT($n$) (C($n$)) molecular SAMs on the Au (111) surface, the $\sqrt{3} \times \sqrt{3}$R30° (Wood's notation) structure of the closed packed molecules with each tilt angle of $\theta = 17°$ (30°) which yields a grafting density $N_o$ (=~4.40 (4.65) × 10$^{18}$ m$^{-2}$ for OPT (alkyl) SAMs) were adopted[28,29]. Using Hertzian elastic-contact model[30] considering the Au-tip radius of ~30 nm (Supplementary Fig. 13), the net force ($P_n$), contact radius ($a$), contact area, grafting density ($N_o$), and the number of each molecular SAMs are estimated and summarized in Supplementary Table 1.

**Pathway-dependent charge transport mechanism.** Because tunneling through the molecular barrier is the dominant charge transport mechanism at a relatively high positive voltage (e.g., $V \approx 1.0$ V), $J$ could be extracted by applying the following tunneling equation[31]:

$$J = \frac{q}{4\pi^2 \hbar d_{tot}^2} \left[ \left( \Phi - \frac{qV}{2} \right) \exp\left( -\frac{2d_{tot}\sqrt{2m}}{\hbar} \sqrt{\Phi - \frac{qV}{2}} \right) - \left( \Phi + \frac{qV}{2} \right) \exp\left( -\frac{2d_{tot}\sqrt{2m}}{\hbar} \sqrt{\Phi + \frac{qV}{2}} \right) \right] \quad (1)$$

where $d_{tot}$ is the total tunneling width (the molecular length in this case), $m$ is the mass of the majority carrier, and $\Phi$ is the molecular barrier height. On the other hand, at relatively high negative voltage (e.g., $V = -1.0$ V), charge transport could occur by both tunneling and Schottky emission across each molecular and MoS$_2$ barrier. In that case, the transport $J$ can be extracted by applying the following equation[32]:

$$J = A^* \cdot T \exp\left[ -\frac{q}{k_B T} \left( \Phi_{SB,eff} - \frac{V}{n} \right) \right] \cdot \exp(\beta \cdot d_m) \quad (2)$$

where $A^*$ is the Richardson constant, $T$ is the temperature, $k_B$ is the Boltzmann constant, $\Phi_{SB,eff}$ is the effective Schottky barrier height for $N_L$-MoS$_2$/Au, $n$ is the ideality factor for MoS$_2$, $\beta$ is the attenuation factor for the molecules, and $d_m$ is the molecular length. Note that both charge transport mechanisms at negative voltage could sequentially occur in the present molecular heterojunction system. $J$ is also estimated in the low voltage regime where charge transport could have sequentially occurred via two tunnel barriers (i.e., the molecular and $N_L$-MoS$_2$ barriers). In such a case, $d_{tot} = d_m + d_{MoS_2}$ and the effective $\Phi$ could be extracted by using the multi-barrier tunneling model[33]:

$$\Phi = \frac{\hbar}{2(2m)^{1/2}} \frac{\frac{2(2m)^{1/2}}{\hbar}(\Phi_m)^{1/2} d_m + \frac{2(2m)^{1/2}}{\hbar}(\Phi_{MoS_2})^{1/2} d_{MoS_2}}{d_m + d_{MoS_2}} \quad (3)$$

where $d_m$ ($d_{MoS_2}$) is the molecular length ($N_L$-MoS$_2$ thickness) and $\Phi_m$ ($\Phi_{MoS_2}$) is the molecular barrier ($N_L$-MoS$_2$ barrier). However, because the interfacial barrier between the molecular SAMs and $N_L$-MoS$_2$ can change depending on the applied voltage polarity, variation of $\Phi$ may be possible (Supplementary Fig. 11). Based on this pathway-dependent charge transport mechanism according to the voltage

polarity, the electrical characteristics for the developed molecular heterojunction systems are theoretically modeled (Fig. 4 and Supplementary Fig. 12). Additional schematics of the pathway-dependent charge transport according to the voltage polarity are presented in the Supplementary Information (Supplementary Fig. 11).

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

## Acknowledgements

This work was accomplished with financial support from the National Research Foundation of Korea (grant no. NRF-2019R1A2C2003704 and 2017R1A5A1014862 (SRC Program: vdWMRC Center)), Korea University Grant, KU-KIST Research Fund, and Korea TORAY Science Foundation. Y.J. and T.L. thank the financial support from the National Creative Research Laboratory program (grant no. 2012026372).

## Author contributions

G.W. and C.-H.L. conceived the project and the idea, J.S and S.Y. designed, carried out, and analyzed the experiments, Y.J. contributed to the temperature-dependent electrical measurement, J.S.E. contributed to the TMD sample preparation, T.-W.K. and T.L. contributed to the data analysis, and all authors analyzed the experimental results and wrote the manuscript.

## Competing interests

The authors declare no competing interests.
