## [Peer Review File · Nature Communications]

Reviewers' comments:

Reviewer #1 (Remarks to the Author):

In this report, Shin et al. present a facile approach to realize a tailored diode based on non-functionalized alkyl and conjugated molecular monolayers that were sandwiched between two-dimensional (2D) semiconductors (MoS₂ and WSe₂) and Au electrodes. By adjusting band alignment at the molecule/2D semiconductor interface, the rectifying characteristics can be implemented and controlled with on/off ratio up to 3.59×10^3 . Depending on the novelty and importance of the work, I would like to recommend its publication after revisions subject to the following comments.

- 1, To demonstrate the reproducibility, the authors should measure the rectification properties with different samples, not only on the different positions of the same sample.
- 2, Since the authors used AFM-based molecular junctions, they might quantify how many molecules they really measured in each experiment.
- 3, One advantage of AFM-based molecular junctions is the capability of carrying out thousands of measurements. This referee suggests the authors to take this advantage to obtain better statistical data to improve the histograms, such as Figures 3b and 3d.

Reviewer #2 (Remarks to the Author):

The experimental finding is interesting. This manuscript may deserve publication in Nature Communications, if the following worries could be resolved:

- 1) The abstract and the introduction do not represent the point of this work. I am afraid that busy readers may miss the point. They might quit going forward to the main body of this paper. Too much emphasis on the novelty of this result of 'heterostructure' origin over previous results of molecular asymmetry origin for the unimolecular rectification is misleading: it sounds like a claim that junctions with asymmetric two electrodes give diode behavior, which is evident. While the band engineering using the heterostructure plays an important role in the present work, it must be explained in an appropriate way.
- 2) The rectification difference between the two kinds of junctions with MoS₂ and WSe₂ monolayer 'coated' substrates is one of the most important results in this manuscript, which is explained in terms of the band diagrams given in Figures 2 and S6. Justification for the band diagram models for the two cases with MoS₂ and WSe₂ is insufficient in spite of some citations. The authors are required to present the best evidence supporting the band diagram models, especially the mid-gap pinning mechanism for the WSe₂ case. While the theoretical model for pathway-dependent charge transport derived from the band diagram seems like to give a fair sketch of experimental results for the MoS₂ case, direct evidences supporting the models itself are crucially important.
- 3) Likewise, the unpinning mechanism suggested for a reason of Figures 3c and 3d is not evidenced enough. The MIGS scenario should be proved more closely. Or the authors may want to present the statement as a speculative discussion. In such a case, the whole presentation should be changed appropriately in such a way like results and discussions.

A successful revision clarifying all the comments, criticisms and requirements stated above will enables me to recommend this manuscript for publication.

Reviewers' comments:

Reviewer #1 (Remarks to the Author):

In this report, Shin et al. present a facile approach to realize a tailored diode based on non-functionalized alkyl and conjugated molecular monolayers that were sandwiched between two-dimensional (2D) semiconductors (MoS_2 and WSe_2) and Au electrodes. By adjusting band alignment at the molecule/2D semiconductor interface, the rectifying characteristics can be implemented and controlled with on/off ratio up to 3.59×10^3 . Depending on the novelty and importance of the work, I would like to recommend its publication after revisions subject to the following comments.

RESPONSE: Thank you for clearly summarizing the main results of our work and for recognizing its novelty. We also appreciate your positive comments about our manuscript and your recommendation that it be published. We have properly revised our manuscript according to your comments and suggestions.

#1. To demonstrate the reproducibility, the authors should measure the rectification properties with different samples, not only on the different positions of the same sample.

#3. One advantage of AFM-based molecular junctions is the capability of carrying out thousands of measurements. This referee suggests the authors to take this advantage to obtain better statistical data to improve the histograms, such as Figures 3b and 3d.

RESPONSE: Thank you for your valuable comments. Because both comments of #1 and #3 are related to reproducibility and statistical analysis of our molecular heterojunction structure, so we address them together.

According to the reviewer's suggestion, we have additionally carried out 1,000 I - V measurements of the Au/OPT2/1_L-MoS₂/Au and Au/C12/1_L-MoS₂/Au with "3" different 1_L-MoS₂ samples, and 200 I - V measurements of Au/OPT2/1_L-WSe₂/Au with another 1_L-WSe₂ samples, respectively (Figs. R1-R4). At least 5 different positions of the same sample were selected for performing the I - V measurements. More statistically meaningful RR values for both OPT2 (Fig. R1) and C12 (Fig. R3) junctions with 1_L-MoS₂ and 1_L-WSe₂ were obtained. We believe this statistical measurement and analysis would be useful for reliably characterizing the electrical features of our molecular heterojunction structure as the reviewer suggested.

Fig. R1. b, Contour maps of transport I for OPT2/1_L-WSe₂ (left) and OPT2/1_L-MoS₂ (right) junctions according to V and the number of junctions. **c,** Statistical histograms of RR for OPT2/1_L-MoS₂ and OPT2/1_L-WSe₂ junctions. The line curves are fitting results from Gaussian function. Note the total numbers of OPT2/1_L-WSe₂ and OPT2/1_L-MoS₂ junctions are 200 and 1,100, respectively.

Fig. R2. (a, d, g, j), Topological line-profiles of different 1_L-MoS_2 samples (#2 (black), #3 (red), and #4 (blue)) and 1_L-WSe_2 (cyan) on SiO_2/Si substrate, investigated in non-contact AFM scanning mode. Black arrows in optical image (inset) indicate the investigation range of the line profiles, respectively. (b, e, h, k), Raman spectra of 1_L-MoS_2 samples (#2 (black), #3 (red), and #4 (blue)) and 1_L-WSe_2 (cyan). (c, f, i, l), Photoluminescence spectra of 1_L-MoS_2 samples (#2 (black), #3 (red), and #4 (blue)) and 1_L-WSe_2 (cyan).

Fig. R3. **a**, Contour maps of transport I for C12/1_L-MoS₂ junctions according to V and the number of junctions. **b**, Statistical histogram of RR for C12/1_L-MoS₂ junction. The line curves are fitting results from Gaussian function. Note the total number of C12/1_L-MoS₂ junctions is 1,100.

Fig. R4. **(a, b)**, RR plots of OPT2/1_L-MoS₂ and C12/1_L-MoS₂ junctions as a function of the sample numbers (#1-#4). Right graphs of **a** and **b** show the statistical histograms of RR for OPT2/1_L-MoS₂ junctions according to the sample numbers (#1-#4).

We have changed Figs. R1b and R1c as Fig. 2b and 2c in the revised manuscript, and added new Figs. R2, R3, and R4 as Figs. S2, S3, and S4 in the revised Supplementary Information (SI). We revised the following paragraph in the revised manuscript (pages 2, 5, 6, and 8) and marked it in blue.

- The rectification ratio could be widely tuned from 1.24 to 1.83×10^4 by changing the molecular species and type and the number of layers of the 2D semiconductors in the heterostructure molecular junction.
- Such rectifying behavior was reproducibly obtained for both molecular heterojunctions through statistical investigation of the I-V characteristics on several positions (more than five positions) at different samples (more than four samples) (200 to 1,100 times), as shown in the contour plots in Fig. 2b and Supplementary Figs. S2-S4. The statistical histogram of RR for the OPT2/1L-MoS₂ junction shows a higher value ($RR = (1.38 \pm 0.73) \times 10^3$) than that for the OPT2/1L-WSe₂ junction ($RR = 2.46 \pm 1.42$) (Fig. 2c). Note that the maximum RR for the OPT2/1L-MoS₂ junction was found to be $\sim 1.83 \times 10^4$ (Fig. S4).
- Note that the statistical data for molecular SAMs/1L-MoS₂ junctions in Figure 3 are obtained from 1L-MoS₂ (Sample #1).

#2. Since the authors used AFM-based molecular junctions, they might quantify how many molecules they really measured in each experiment.

RESPONSE: We estimated the number of the participated molecules in our molecular heterostructure based on *Hertzian elastic contact model*^[1]. The Hertzian elastic contact model can be utilized to estimate the contact radius (a) between Au tip and molecular SAMs, $a = (RP_n/K)^{1/3}$, where R is the Au-tip radius (~ 30 nm in Fig. R5), P_n is the net force determined by F_L (tip-loading force) + F_{adhesion} (adhesion force), and K is elastic modulation (~ 20 GPa)^[2]. Note that it was found $F_{\text{adhesion}} = \sim 31.6$ nN, 33.6 nN, ~ 13.2 nN, 15.4 nN, and 16.2 nN for OPT1, OPT2, C8, C10, and C12, respectively.^{[2], [3]} To estimate the number of molecules on Au tip, we adopted the well-known ordered $\sqrt{3} \times \sqrt{3}R30^\circ$ (Wood's notation) structure of the closed packed molecules with a tilt angle of $\theta = 17^\circ$ and a grafting density of $N_o \approx 4.40 \times 10^{18} \text{ m}^{-2}$ in OPT groups and $\sqrt{3} \times \sqrt{3}R30^\circ$ structure of the closed packed molecules with a tilt angle of $\theta = 30^\circ$ and a grafting density of $N_o \approx 4.65 \times 10^{18} \text{ m}^{-2}$ in alkyl groups.^{[4], [5]}

Based on this model, we summarized P_n , contact radius (a), contact area, grafting density (N_o), and the # of each molecular SAMs in Table R1.

	F_L (nN)	P_n (nN)	a (nm)	Contact area (nm ²)	N_o (nm ⁻²)	# of molecules
OPT1	1	31.6	3.62	41.14	4.4	181
OPT2	1	34.6	3.73	43.71	4.4	192
C8	1	14.2	2.77	24.14	4.65	112
C10	1	16.4	2.91	26.57	4.65	123
C12	1	17.2	2.96	27.43	4.65	127

Table R1. Summary for P_n , a , contact area, N_o , and the # of molecules of each molecular SAMs.

Fig. R5. SEM image of a CAFM Au probe tip.

Reference

- [1] Johnson, K. L. Contact Mechanics; *Cambridge University Press: New York*, 1985; pp 104-106
- [2] Song, H., Lee, H. & Lee, T. Intermolecular chain-to-chain tunneling in metal-alkanethiol-metal junctions. *J. Am. Chem. Soc.* **129**, 3806-3807 (2007).
- [3] Shin, J., Gu, K., Yang, S., Lee, C.-H., Lee, T., Jang, Y. H. & Wang, G. Correlational effects of the molecular-tilt configuration and the intermolecular van der Waals interaction on the charge transport in the molecular junction. *Nano Lett.* **18**, 4322-4330 (2018).
- [4] Sabatani, E., Cohen-Boulakia, J., Bruening, M. & Rubinstein, I. Aromatic monolayers on gold: A new family of self-assembling monolayers. *Langmuir* **9**, 2974-2981 (1993).
- [5] Poirier, G. E. & Tarlow, M. J. The $c(4\times 2)$ superlattice of n -alkanethiol monolayers self-assembled on Au(111). *Langmuir* **10**, 2853-2856 (1994).

In response to your comment, we have added Fig. R5 and Table R1 in the revised SI. Also, we have added the following paragraph on page 12 in revised Methods section and on the page 17 in the revised SI.

- *Estimated Number of Contacted Molecules*
For the OPT(n) (C(n)) molecular SAMs on the Au (111) surface, the $\sqrt{3} \times \sqrt{3}R30^\circ$ (Wood's notation) structure of the closed packed molecules with each tilt angle of $\theta = 17^\circ$ (30°) which yields a grafting density N_o ($= \sim 4.40$ (4.65) $\times 10^{18} \text{ m}^{-2}$ for OPT (alkyl) SAMs) were adopted.^{28,29} Using Hertzian elastic-contact model³⁰ considering the Au-tip radius of ~ 30 nm (Fig. S13), the net force (P_n), contact radius (a), contact area, grafting density (N_o), and the number of each molecular SAMs were estimated and summarized in Table S1.
- *The Hertzian elastic contact model can be utilized to estimate the contact radius (a) between the Au tip and SAMs, $a = (RP_n/K)^{1/3}$, where R is the Au-tip radius ($R = \sim 30$ nm, Fig. S13), P_n is the net force determined by F_L (tip loading force) + $F_{adhesion}$ (adhesion force), and K is elastic modulation (~ 20 GPa).¹⁰ Note that it was found $F_{adhesion} = \sim 31.6$ nN, 33.6 nN, ~ 13.2 nN, 15.4 nN, and 16.2 nN for OPT1, OPT2, C8, C10, and C12, respectively.^{5,10} Based on these parameters, the contact radius, contact area, and the number of each molecular SAMs are estimated and summarized in Table S1.*

In summary, we have revised and improved the manuscript and Supplementary Information according to all your comments. We hope that the quality of our work now satisfies the requirements for publication in *Nature Communications*, and we believe our work will be of interest to readers and researchers in the field of molecular electronics and 2D electronics. We sincerely appreciate you taking the time to evaluate our revised manuscript. Please note that we have added new co-author, Jung Sun Eo, who contributed to the TMD sample preparation (1_L -MoS₂ and 1_L -WSe₂) for the revised manuscript and Supplementary Information. Thank you again.

Reviewers' comments:

Reviewer #2 (Remarks to the Author):

The experimental finding is interesting. This manuscript may deserve publication in Nature Communications, if the following worries could be resolved:

RESPONSE: Thank you for evaluating our work is interesting. We also really appreciate your positive comments about our manuscript and your recommendation for its publication. We have revised our manuscript according to your comments and suggestions.

#1. The abstract and the introduction do not represent the point of this work. I am afraid that busy readers may miss the point. They might quit going forward to the main body of this paper. Too much emphasis on the novelty of this result of 'heterostructure' origin over previous results of molecular asymmetry origin for the unimolecular rectification is misleading: it sounds like a claim that junctions with asymmetric two electrodes give diode behavior, which is evident. While the band engineering using the heterostructure plays an important role in the present work, it must be explained in an appropriate way.

RESPONSE: Thank you for your kind comment and suggestion. And we agree with your comment on “*While the band engineering using the heterostructure plays an important role in the present work, it must be explained in an appropriate way.*”.

As the reviewer suggested, we have properly revised the Introduction/Conclusion parts. Especially, we have further emphasized the band engineering between molecules and 2D materials that can create a rectifying feature than heterostructure or asymmetric contact itself.

We revised following paragraph in the revised manuscript (page 3, 4, 10, and 11) and marked it in blue.

- *The ultimate objectives in the field of molecular electronics are to realize electronic functionalities, such as rectifying, optical switching, and thermoelectric effects, at the device miniaturization limit and to systemize all the aspects of the charge transport mechanisms for rational device design.¹⁻¹² Specially designed molecular species have been utilized for the realization of specific (or desired) device functions. For example, the D- σ -A type molecules have been widely used for a diode component, where the highest-occupied molecular orbital (HOMO) of the donor and the lowest-unoccupied molecular orbital (LUMO) of the acceptor are closely aligned to the Fermi level of electrodes, i.e., the frontier orbital level of each molecular unit is asymmetrically positioned with respect to the electrodes.^{9,10} This asymmetric energy alignment in the junction can cause the different sequential tunneling through the acceptor and donor depending on the polarity of the applied voltage, which could yield the rectification property of the molecular junction. In the case of the ferrocenyl molecules that composed of long alkyl chain and ferrocenyl terminus unit, they can also exhibit the rectification characteristic because the charge transport pathways are mainly determined by the relative HOMO level of the ferrocenyl unit according to voltage polarities.^{11,12} In this sense, specially designed molecular species that can differently engineer the band alignment of the molecular units according to the voltage polarity is dispensable for implementing a desirable diode's behavior in molecular electronic junction.*
- *In this point of view, non-functionalized molecules such as alkanethiol or conjugated molecules that only exhibit symmetric tunneling transport have been excluded from the realization of electronic diode functions.^{13,14} However, even these widely studied molecules*

can present the desired electronic functionality as the interfacial band alignment is properly adjusted in molecular heterojunctions. Here, we present a novel strategy and design rule for realizing molecular-scale diode features based on the energy band engineering between simple alkanethiol or conjugated molecules and 2D semiconductors. We systematically engineered the band alignment by inserting 2D semiconductors between molecules and metal (Au), resulting in different charge transport pathways according to voltage polarities without introducing specially designed molecules. In addition, the rectifying characteristics are tunable by controlling the essential constituents such as the molecular species, molecular length, 2D semiconductors, and the number of layers. Our work sets a generalized design rule for implementing rectifying characteristics in a molecular heterojunction structure with non-functionalized molecules and 2D semiconductors.

- In summary, we have first implemented a molecular rectifier composed of simple molecules and 2D semiconductors. The rectifying characteristics can be implemented and further tuned by engineering band alignment between non-functional molecules and 2D semiconductors which make different transport pathways according to voltage polarities. Given that non-functional SAM-based molecular heterojunctions employing 2D semiconductors exhibit strong rectifying characteristics and that their performance can be controlled by implementing the suggested design rule, a tailored molecular rectifier based on simple σ - and π -bonded SAMs can potentially be realized.

#2. The rectification difference between the two kinds of junctions with MoS₂ and WSe₂ monolayer ‘coated’ substrates is one of the most important results in this manuscript, which is explained in terms of the band diagrams given in Figures 2 and S6. Justification for the band diagram models for the two cases with MoS₂ and WSe₂ is insufficient in spite of some citations. The authors are required to present the best evidence supporting the band diagram models, especially the mid-gap pinning mechanism for the WSe₂ case. While the theoretical model for pathway-dependent charge transport derived from the band diagram seems like to give a fair sketch of experimental results for the MoS₂ case, direct evidences supporting the models itself are crucially important.

RESPONSE: We appreciate your comment and fully recognize the importance of the following comment “The authors are required to present the best evidence supporting the band diagram models, especially the mid-gap pinning mechanism for the WSe₂ case.” and “direct evidence supporting the models itself are crucially important.”

In order to further investigate the Fermi-level pinning behavior at the metal/WSe₂ interface, we have additionally conducted the I - V measurements for the OPT2/1_L-WSe₂ junction using another bottom electrode (Pt) in addition to Au. As shown in Fig. R2, the electrical characteristic and the extracted RR value were rarely changed as compared with those of the OPT2/1_L-WSe₂ junction using the Au electrode. This is presumably because the Fermi-level of the metal is similarly pinned at the mid-gap states of 1_L-WSe₂ regardless of the metal work functions (Au (~5.1 eV) and Pt (5.7 eV)) although the Fermi levels of both metals can align relatively well with the valence band edge of WSe₂ only if considering their work functions.^[1] According to previous literature, it has been well known that the midgap states can be formed by both defect states originated from naturally existing vacancies of WSe₂ and virtual gap states induced by metal wave function penetration, which results in the strong Fermi level pinning.^[2] We believe that our control experimental results could support our energy-band diagram model for the Au/OPT2/1_L-WSe₂/Au junction that was based on the mid-gap pinning for the WSe₂ case (Fig. S7 in the revised Supplementary Information).

Of course, we are fully aware that this experiment scheme is not sufficient for the direct evidence of the mid-gap pinning. We believe that the analysis of gate-dependent electrical characteristics could give more direct evidence for where the Fermi-level of the electrode is aligned into the 1_L-WSe₂. However, this three-terminal experiment is beyond present our experimental design and capability. As an alternative way, various metals with low work functions, such as Ag, Ti and Cr, can be employed to investigate molecular junction characteristics. But, it is also difficult because such metals are easily

oxidized under ambient experimental conditions. The related discussion is added in the revised manuscript.

Fig. R1. **a**, Topological line-profile of 1_L -WSe₂ samples on Pt substrate, investigated by non-contact AFM scanning mode. Black arrow in optical image (inset) indicates the investigation range of the line profiles, respectively. **b**, Raman spectra of 1_L -WSe₂ samples. **c**, Photoluminescence spectra of 1_L -WSe₂ samples.

Fig. R2. *RR* plots of Au/OPT2/ 1_L -WSe₂/Au (black) and Au/OPT2/ 1_L -WSe₂/Pt (red), respectively (left). The statistical histograms of *RR* for Au/OPT2/ 1_L -WSe₂/Au (black) and Au/OPT2/ 1_L -WSe₂/Pt (red), respectively (right).

Reference

[1] Chiu, M.-H., Tseng, W.-H., Tang, H.-L., Chang, Y.-H., Chen, C.-H., Hsu, W.-T., Chang, W.-H., Wu, C.-I. & Li, L.-J. Band alignment of 2D transition metal dichalcogenide heterojunctions. *Adv. Funct. Mater.*, **27**, 1603756 (2017).

[2] Guo, Y., Liu, D. & Robertson, J. Chalcogen vacancies in monolayer transition metal dichalcogenides and Fermi level pinning at contacts. *Appl. Phys. Lett.* **106**, 173106 (2015).

In this regard, we have added following paragraph in the revised manuscript (page 7) and Figs. R1 and R2 as Figs. S6 and S7 and marked in blue.

- *In order to further investigate the Fermi-level pinning behavior at the metal/WSe₂ interface, we have additionally conducted the I-V measurements for the OPT2/ 1_L -WSe₂ junction using another bottom electrode (Pt) in addition to Au. The electrical characteristic and the extracted *RR* value were rarely changed as compared with those of the OPT2/ 1_L -WSe₂ junction using the Au electrode (Supplementary Figs. S6 and S7). This is presumably because the Fermi-level of the metal is similarly pinned at the mid-gap states of 1_L -WSe₂ regardless of the metal work functions (Au (~5.1 eV) and Pt (5.7 eV)) although the Fermi levels of both metals can align relatively well with the valance band edge of WSe₂ only if considering their work functions.²³ According to previous literature, it has been well known that the midgap states can be formed by both defect states originated from naturally existing*

vacancies of WSe₂ and virtual gap states induced by metal wave function penetration, which results in the strong Fermi level pinning.²² Therefore, our control experimental results support our energy-band diagram model for the Au/OPT2/1_L-WSe₂/Au junction that was based on the mid-gap pinning for the WSe₂ case. It is noted that an additional experiment such as the analysis of gate-dependent electrical characteristics is desired for the direct evidence for where the Fermi-level of the electrode is aligned into the 1_L-WSe₂.

#3. *Likewise, the unpinning mechanism suggested for a reason of Figures 3c and 3d is not evidenced enough. The MIGS scenario should be proved more closely. Or the authors may want to present the statement as a speculative discussion. In such a case, the whole presentation should be changed appropriately in such a way like results and discussions.*

RESPONSE: Thank you for your valuable comment.

Generally, there are two main origins of interface gap states that cause the Fermi level pinning at the metal/semiconductor interface; 1) defect-induced gap states (DIGS) and 2) metal-induced gap states (MIGS). Among them, the MIGS can be significantly reduced by adjusting the interfacial gap distance. According to previous studies for 2D semiconductors as well as 3D material systems, it is evident that the insertion of insulating layers between the metal and the semiconductor alleviates the Fermi-level pinning effect by reduction of the MIGS density in semiconductors. So, it can be inferred that the molecules as an insulating layer between the Au probe tip and TMD layers can alleviate the Fermi-level pinning effects in our molecular heterojunction structures. The degree of Fermi-level depinning effect in molecules can be different from that in inorganic insulating layers such as TiO₂ or Al₂O₃, however, the relation between MIGS density and insulating layer thickness is presumably similar regardless of insulating layer species. Despite such consideration, we cannot prove the MIGS scenario more directly for our molecular junction case. Therefore, we have toned down MIGS statements as a speculative discussion for Figs. 3c and 3d.

In response to your comment, we have revised the statements as a speculative discussion for Figs. 3c and 3d in the revised manuscript. We have added the following paragraph on page 9 in revised manuscript.

- *Inserting the thicker insulating layer between the metal and the semiconductor can further alleviate the E_F pinning effects by reduction of the metal-induced gap state (MIGS) originating from further attenuation of the charge wave function across the insulating layer.^{26,27} In this regard, inserting the longer molecules acting as an insulating layer between the Au-tip and 1_L-MoS₂ might lead to further unpinning of the E_F by reduction of MIGS density in 1_L-MoS₂.*

Reference

- [1] Wager, J. F. & Robertson, J. Metal-induced gap states modeling of metal-Ge contacts with and without a silicon nitride ultrathin interfacial layer. *J. Appl. Phys.* **109**, 094501 (2011).
- [2] Kim, G.-S. et al. Schottky barrier height engineering for electrical contacts of multilayered MoS₂ transistors with reduction of metal-induced gap states. *ACS Nano* **12**, 6292-6300 (2018).

In summary, we have revised and improved the Introduction and statements for Fermi level pinning effects in the manuscript. We hope that the quality of our work now satisfies the requirements for publication in *Nature Communications*, and we believe our work will be of interest to readers and researchers in the field of molecular electronics and 2D electronics. We sincerely appreciate you taking the time to evaluate our revised manuscript. Please note that we have added new co-author, Jung Sun Eo, who contributed to the TMD sample preparation (1_L-MoS₂ and 1_L-WSe₂) for the revised manuscript and Supplementary Information.

Thank you again.

REVIEWERS' COMMENTS:

Reviewer #1 (Remarks to the Author):

The authors have answered my comments properly. So, I recommend its publication.

Reviewer #2 (Remarks to the Author):

In this revision, the authors have clarified the benefit and the limitation of their achievement. I believe that this version of the manuscript will be useful to readers of Nature Communications. I recommend it for the publication.